# Mutational Profile of Malignant Pleural Mesothelioma (MPM) in the Phase II RAMES Study

**DOI:** 10.3390/cancers12102948

**Published:** 2020-10-13

**Authors:** Maria Pagano, Luca Giovanni Ceresoli, Paolo Andrea Zucali, Giulia Pasello, Marina Garassino, Federica Grosso, Marcello Tiseo, Hector Soto Parra, Francesca Zanelli, Federico Cappuzzo, Francesco Grossi, Filippo De Marinis, Paolo Pedrazzoli, Roberta Gnoni, Candida Bonelli, Federica Torricelli, Alessia Ciarrocchi, Nicola Normanno, Carmine Pinto

**Affiliations:** 1Oncology Unit, Clinical Cancer Centrer, Azienda Unità Sanitaria Locale (AUSL)-IRCCS di Reggio Emilia, 42121 Reggio Emilia, Italy; zanelli.francesca@ausl.re.it (F.Z.); gnoni.roberta@ausl.re.it (R.G.); bonelli.candida@ausl.re.it (C.B.); pinto.carmine@ausl.re.it (C.P.); 2Medical Oncology, Cliniche Humanitas Gavazzeni, 42121 Bergamo, Italy; giovanni_luca.ceresoli@gavazzeni.it; 3Department of Oncology, Humanitas Clinical and Research Center, IRCCS, 42557 Rozzano, Italy; paolo.zucali@hunimed.eu; 4Department of Biomedical Sciences, Humanitas University, 20090 Pieve Emanuele, Italy; 5Department of Oncology, Medical Oncology 2 Istituto Oncologico Veneto IRCCS, 35128 Padova, Italy; giulia.pasello@ioveneto.it; 6Department of Medical Oncology Fondazione IRCCS Istituto Nazionale dei Tumori, 20090 Milan, Italy; garassino.studiclinici@istitutotumori.mi.it; 7Azienda Ospedaliera SS Antonio e Biagio e Cesare Arrigo, Mesothelioma and rare cancer unit, 15121 Alessandria, Italy; federica.grosso@ospedale.al.it; 8Department of Medicine and Surgery, University of Parma, 43121 Parma, Italy; mtiseo@ao.pr.it; 9Medical Oncology Unit, University Hospital of Parma, 43121 Parma, Italy; 10Department of Oncology, Medical Oncology, University Hospital Policlinico-San Marco, 95100 Catania, Italy; hsotoparra@yahoo.it; 11UOC Oncologia Medica 2 Istituto Nazionale Tumori Regina Elena, 00100 Roma, Italy; Federico.cappuzzo@auslromagna.it; 12UOC Oncologia Medica Fondazione IRCCS Ca’ Granda Ospedale Maggiore Policlinico, 20090 Milan, Italy; francesco.grossi@policlinico.mi.it; 13European Institute of Oncology, IRCCS, 20090 Milan, Italy; Filippo.DeMarinis@ieo.it; 14Fondazione IRCCS Policlinico San Matteo, 27100 Pavia, Italy; p.pedrazzoli@smatteo.pv.it; 15Department of Internal Medicine and Medical Therapy, University of Pavia, 27100 Pavia, Italy; 16Laboratory of Translational Research, Azienda Unità Sanitaria Locale-IRCCS di Reggio Emilia, 42100 Reggio Emilia, Italy; federica.torricelli@ausl.re.it (F.T.); Alessia.Ciarrocchi@ausl.re.it (A.C.); 17Cell Biology and Biotherapy Unit, Istituto Nazionale Tumori, “Fondazione G. Pascale”-IRCCS, 80100 Napoli, Italy; n.normanno@istitutotumori.na.it

**Keywords:** malignant pleural mesothelioma, gene expression profile, *BAP1*, chemotherapy

## Abstract

**Simple Summary:**

Malignant pleura Mesothelioma (MPM) is an aggressive cancer arising from the mesothelial cells of the pleura. About 80% of mesothelioma cases are linked to asbestos exposure; the remainder may be related to prior chest radiation, genetic predisposition or spontaneous occurrence. Understanding the genetic alterations that drive MPM is critical for successful development of diagnostics, prognostics and personalized therapeutic modalities. Because MPM is rare, genomic studies are limited and have typically involved a small number of samples. The aim of our study was to understand the mutational landscape of MPM. Our analysis identified significantly mutated genes. This study on a rare tumor type will be important for patients “in real life”.

**Abstract:**

*Purpose:* Malignant pleural mesothelioma (MPM) is an aggressive cancer. Data are not available in prospective trials on correlations between genetic alterations and outcomes of therapies. In this study, we assessed the genetic profile of MPM patients (pts) in tissue samples. *Patients and Methods:* From December 2016 to July 2018 (end of enrolment), 164 pts were enrolled. We evaluated by targeted sequencing the mutational profile of a panel of 34 genes: *ACTB*, *ACTG1*, *ACTG2*, *ACTR1A*, *BAP1*, *CDH8*, *CDK4*, *CDKN2A*, *CDKN2B*, *COL3A1*, *COL5A2*, *CUL1*, *DHFR*, *GOT1*, *KDR*, *KIT*, *MXRA5*, *NF2*, *NFRKB*, *NKX6-2*, *NOD2*, *PCBD2*, *PDZK1IP1*, *PIK3CA*, *PIK3CB*, *PSMD13*, *RAPGEF6*, *RDX*, *SETDB1*, *TAOK1*, *TP53*, *TXNRD1*, *UQCRC1*, *XRCC6.* Genetic profiling was correlated with clinical and pathological variables. *Results:* Overall, 110 pts (67%) from both treatment arms had samples available for molecular analysis. Median age was 63 years (45–81), 25.5% (*n* = 28) were females, and 74.5% (*n* = 82) were males. Tumor histotype was 81.8% (*n* = 90) epithelioid and 18.2% (*n* = 20) non-epithelioid; 28.5% of the tumors (*n* = 42) were stage IV, 71.5% (*n* = 68) were stage III. Targeted sequencing of tissue specimens identified 275 functional somatic mutations in the 34 genes analyzed. The number of mutated genes was positively associated with higher stage and metastatic disease (*p* = 0.025). *RDX* (42%), *MXRA5* (23%), *BAP1* (14%), and *NF2* (11%) were the most frequently mutated genes. Mutations in *RAPGEF6* (*p* = 0.03) and *ACTG1* (*p* = 0.02) were associated with the non-epithelioid subtype, and mutations in *BAP1* (*p* = 0.04) were related to progression-free survival (PFS) > 6 months. *Conclusions:* In the Ramucirumab Mesothelioma clinical trial (RAMES), mutation of the gene *BAP1* is related to a prolonged PFS for patients treated with platinum/pemetrexed regimens (*p* = 0.04).

## 1. Introduction

Malignant pleural mesothelioma (MPM) is a rare, highly aggressive malignancy of the mesothelium that is usually diagnosed at an advanced stage. Chemotherapy, which is the only available therapeutic option in the clinical setting, is modestly effective in MPM [1].

The proposed prognostic factors include clinical variables, radiological parameters at presentation, and pathological/molecular findings; however, the vast majority of these factors are not fully validated [2] Histologic type remains one of the most reliable prognostic factors, as the epithelioid subtype is associated with a better prognosis, and the sarcomatoid subtype with the worst one. Recently, thymidylate synthase expression has been proposed as a predictor of sensitivity to pemetrexed [3,4].

MPM is a cancer caused by occupational and environmental exposure to asbestos fibers [5]. A very rare condition is susceptibility to MPM in some families [6]. The subsequent discovery of an MPM risk in family members who are heterozygous for inherited/germline *BAP1* mutations underscores the role of genetic alterations in this disease [7].

Because MPM is a rare tumor, genomic studies are limited and have often involved a small number of samples. No data are available in prospective clinical trials. Loss-of-function mutations in the cyclin-dependent kinase inhibitor 2A gene (CDKN2A), the neurofibromin 2 gene (*NF2*), and the *BRCA1*-associated protein-1 gene (*BAP1*) have been reported in MPM [8,9]. In addition, previous studies have reported copy gains and copy losses involving multiple regions of the genome.

We report the results of the genomic mutational profile of 110 patients enrolled in the Ramucirumab Mesothelioma clinical trial (RAMES) and the correlation with outcomes of platinum/pemetrexed first-line chemotherapy. 

## 2. Material and Methods 

The RAMES Study (EudraCT number 2016-001132-36) is a multicenter, double-blind, randomized Phase II trial exploring the efficacy and safety of the addition of ramucirumab to gemcitabine as a second-line treatment for patients with malignant pleural mesothelioma. The RAMES Study is being conducted in 26 Italian oncology centers, in accordance with the International Council on Harmonization Good Clinical Practice guidelines and the Declaration of Helsinki (Figure 1).

The patients were assigned (1:1) to receive intravenous gemcitabine 1000 mg/m^2^ on days 1 and 8 every 21 days, with either placebo or intravenous ramucirumab 10 mg/kg on day 1 of a 21-day cycle, until maximum patient tolerability was reached or progressive disease (PD) occurred. Random assignment was stratified by performance status (0–1 vs. 2), age (≤70 vs. >70), histology (epithelioid vs. others), and time to progression (TTP) after a previous treatment. The primary endpoint was overall survival (OS), and the secondary endpoints were progression-free survival (PFS), overall response rate (ORR), safety, quality of life (QoL), and predictive biomarkers.

In the present report, the correlation between the genomic mutational profile of 110 patients enrolled in the RAMES trial and outcomes of the first line of therapy with the platinum/pemetrexed regime is described (PFS1 and ORR).

The PFS1 was collected retrospectively and calculated from the date of the first cycle to radiological progression.

The ORR of each treatment was calculated as defined by RECIST v 1.1 guidelines. Not central revision was performed.

The RAMES trial is ongoing, and OS data are not mature yet. In the final analysis, the OS impact of the second-line treatment and genomic mutational profile will be evaluated. 

Formalin-fixed paraffin-embedded (FFPE) tissue samples were obtained by pleuroscopy or video thoracoscopy at diagnosis from patients with Stage III and IV MPM prior first-line treatment with platinum plus pemetrexed chemotherapy.

A second biopsy was not performed before starting the second line of therapy. 

All samples were reviewed and classified for histotype according to the current WHO classification [9]. After revision of the tumor type and neoplastic cell content from the available FFPE tissue samples, five hematoxylin-and-eosin-stained slides were sent for DNA analysis. Before analyzing the tissue blocks and slides, all samples were de-identified, and all cases were anonymized by a pathology staff member involved in this project. Informed consent was obtained from each patient, and the institutional review board of the institution approved the study (161/2016/FARH/IRCCS/Reggio Emilia, Italy).

A targeted sequencing panel of 1041 amplicons was designed based on literature data, spanning 34 genes frequently involved in MPM tumors (*ACTB*, *ACTG1*, *ACTG2*, *ACTR1A*, *BAP1*, *CDH8*, *CDK4*, *CDKN2A*, *CDKN2B*, *COL3A1*, *COL5A2*, *CUL1*, *DHFR*, *GOT1*, *KDR*, *KIT*, *MXRA5*, *NF2*, *NFRKB*, *NKX6-2*, *NOD2*, *PCBD2*, *PDZK1IP1*, *PIK3CA*, *PIK3CB*, *PSMD13*, *RAPGEF6*, *RDX*, *SETDB1*, *TAOK1*, *TP53*, *TXNRD1*, *UQCRC1*, *XRCC6*). Libraries were prepared using TruSeq Custom Amplicon Low Input Kit (Illumina), and sequencing was performed on NextSeq 500 Mid-Output Reagent Cartridge v2 300 cycles (2 × 151). Primary bioinformatics analysis was conducted using Amplicon DS software included in the Illumina BaseSpace environment. Mutation analysis, annotation, and filtering were performed using VariantStudio software (Illumina, San Diego, CA, USA). 

Mutations were selected based on VCF (Variant Call Format) quality filters excluding strand bias, low-quality, and low-coverage variants. Mutations were considered “passing filter” if covered by at least 1000 reads and if identified with a minimum frequency of 3% (not less than 40 reads containing the variation). Somatic and functional mutations were identified by a bioinformatic selection that excluded from further analyses variants described with a frequency in the ExAC (Exome Aggregation Consortium) general population higher than 0.1% or classified as synonymous and intronic. 

About 75% of the identified variants were present with a frequency lower than 20%, and their validation using more standard but less sensitive techniques such as Sanger sequencing was not feasible. Aware of the importance of variant calling validation, we are in the process of developing new approaches for the validation of the most functionally relevant variants based on more sensitive techniques like Droplet digital PCR. This activity will take a significant set-up time which will fall out of the timing of this manuscript and likely be the subject of future researches.

Mutations were considered reliable if sequenced with a minimum coverage of 1000×. Only mutations with a frequency in the ExAC general population < 0.1%, described as non-synonymous and non-intron, were considered.

## 3. Statistical Analysis

Statistical analysis was performed using the R Software v 3.5.1. “GenVisR” package to generate waterfall plots. Unsupervised hierarchical clustering was created considering only genetic variants. The mutational status of each gene was considered as a categorical variable where the presence/absence of variants were binary codified. Analysis of the frequency of gene mutations in subgroups of patients with different clinical features was performed by applying Fisher’s exact test and Kruskal–Wallis test for categorical and continuous variables, respectively. Differences were considered statistically significant for a *p* value < 0.05.

## 4. Results

From December 2016 to July 2018 (end of enrolment), a total of 164 patients were randomized. In January 2020, 13/164 patients were on treatment, and 100/164 patients had died. At the time of the present report, the number of 114 events required by the statistical design for opening a blind trial has not yet been reached, and efficacy data are not yet available. 

A total of 110/164 tumor tissue samples were available for targeted sequencing analysis. The patients’ characteristics are reported in Table 1**.** The ECOG-Performance Status (PS)was 0–1 in 89% and 2 in 11% of the patients. The median age was 63 years (45–81); 25.5% (*n* = 28) were females, and 74.5% (*n* = 82) were males. The histotype was 81.8% (*n* = 90) epithelioid and 18.2% (*n* = 20) non-epithelioid; 28.5% of the tumors (*n* = 42) were Stage IV, 71.5% (*n* = 68) were Stage III.

The first-line therapy was cisplatin plus pemetrexed or carboplatin plus pemetrexed. The median PFS in the first-line platinum/pemetrexed (PFS1) therapy was 5.75 months (C.I. 95% 4.75–6.76). PFS was ≤6 months in 49% of patients and > 6 months in 41% (10% pts not available). 

The ORR at first-line chemotherapy was partial response in 27.3% (*n* = 30) of patients, stable disease in 32.7% (*n* = 36) (disease control rate 60.0%), and progressive disease in 32.7% (*n* = 36). In 7.3% (*n* = 8) of patients the response was unknown. 

A total of 275 functional somatic mutations were identified in the 34 genes analyzed; 74% were missense variants, 17% were splice variant, 5% were start/stop modification, 3% were frameshift variants, and 1% were in-frame deletions. The vast majority of the detected mutations (90%) were likely to be somatic, with an allelic frequency ranging from 3% to 50%. *RDX* (42%), *MXRA5* (23%), *BAP1* (14%), and *NF2* (11%) were the most frequently mutated genes (Figure 2A). We found that 50% of the patients showed mutations in two to five genes. In 10% of the patients, no genetic mutations were identified in the investigated genes (Figure 2B). 

We found a significant correlation between number of mutations and PFS1 in our cohort. In fact, the average number of somatic mutations in patients with PFS at first-line chemotherapy ≤6 and >6 months was 1.6 and 2.2 (*p* = 0.032), respectively.

The number of mutated genes was positively associated with higher tumor stage (*p* = 0.025).

In our study, 48.2% of the patients had a history of exposure to asbestos. For 4.5% of the patients, asbestos exposure information was not available. No significant differences were found in gene mutations according to the asbestos exposure status. Similarly, we found no correlation between type of gene mutations and disease stage.

Genetic alterations in *RAPGEF6* (*p* = 0.035) and *ACTG1* (*p* = 0.027) were more frequent in the sarcomatoid and biphasic subtype of MPM as compared with the epithelioid subtype (Figure 3).

*BAP1* variants (*p* = 0.04) were associated with a longer PFS after platinum/pemetrexed first-line treatment (PFS1) (Figure 4A,B). In addition, *CUL1* and *DHFR* mutations were related to the epithelioid subtype, Stage III, and PFS1 > 6 months, but this correlation was not statistically significant.

Unsupervised clustering identified four molecular MPM subgroups on the basis of the genetic alterations identified in the gene panel. Cluster 1 was defined by a high incidence of *MXRA5* mutations, cluster 2 included the majority of MPM cases mutated in *BAP1*, *NF2*, *NOD2*, or *RAPGEF6*, cluster 3 was comprised of 91% of MPM cases mutated in *PIK3CB*, and cluster 4 included the majority of MPMs mutated in *RDX* and more than 80% of cases mutated in *ACTG1* (Figure 5). As a whole, the clinicopathological characteristics, including age, sex, asbestos exposure, stage (III and IV), ORR and PFS at the first-line treatment, and histological type, did not have a significantly different distribution among the four groups.

## 5. Discussion

Malignant mesothelioma is an uncommon cancer with limited therapeutic options and poor clinical outcomes. In an effort to identify actionable targets in MPM, several studies have reported on its genomic background. The use of massive parallel sequencing has contributed to identifying the main genetic alterations in this disease. In particular, the most comprehensive genomic analysis to date by Bueno et al. [10] describes the exome and transcriptome sequencing of 216 tumor and control specimens of patients with MPM. Exome analysis revealed that *BAP1*, *NF2*, *TP53*, *SETD2*, *DDX3X*, *ULK2*, *RYR2*, *CFAP45*, *STDB1*, and *DDX51* were the most frequently mutated genes in MPM. Other studies have reported genetic alterations in a number of different genes using either exome sequencing or targeted sequencing approaches [11,12].

In the RAMES study, we took advantage of the results of the above studies to investigate the genomic landscape of 110 MPM tumors using a targeted sequencing panel covering the 34 genes more frequently mutated in MPM. 

We acknowledge that the choice of a restricted panel of genes is one of the limitations of this work and that the use of a wider approach would have permitted a deeper investigation of RAMES patients’ genetic profile. However, previous studies conducted on MPM had already demonstrated that this kind of tumor is characterized by low genetic complexity and by the existence of a few number of recurrently mutated genes. For this reason, we included in the panel only a selection of genes extracted from a careful literature revision to cover the most frequent alterations in MPM and to ensure a statistically relevant representation of these genetic events within our cohort [11].

Importantly, patients in the RAMES study had advanced tumors, not amenable to surgery. Therefore, this is a unique population in which the genetic landscape of MPM can be investigated.

All patients received pemetrexed/platinum-based doublet as first-line treatment, with a median PFS of 5.75 months. There was as usual a majority of tumors of the epithelioid subtype (82%), with a majority of Stage III tumors (71%).

Maturity of survival analysis looks satisfying, since 100 patients/164 already died (61%) at the time of the analysis, although 114 events will be expected for efficacy analysis of the experimental arm. 

We are aware that this study presents some technical limitations. Due to the low allelic frequency of the majority of somatic mutations, their validation using more standard but less sensitive techniques such as Sanger sequencing was not feasible. Moreover, it was not possible to compare tumors with their matched normal counterparts because of limited biological material. To overcome this problem, a bioinformatic selection based on variants effect and reported frequency in the total population was carried out in order to identify somatic alterations.

We found a significant correlation between number of mutations and PFS1 in our cohort. In addition, the number of mutated genes was positively associated with higher tumor stage and metastatic disease. Such correlations might suggest that tumors with higher genomic instability and possibly higher heterogeneity might have a more aggressive behavior. This observation is in agreement with previous reports that showed a correlation between tumor heterogeneity and poor prognosis in cancer patients [13].

Unsupervised clustering analysis based on genetic alterations identified four different groups within our cohort. However, no correlation with clinical and morphological parameters could be identified. MPM is a highly heterogeneous disease in terms of clinical, morphological, and molecular features, whose complexity may not just be explained on the basis of genetic alterations. Our observation, in line whit such complexity, may indicate that the identified clusters are just reflection of this heterogeneity. Integration with additional molecular profiling could help to enhance the meaning of this clusterization and its clinical relevance. The strong point of such study is that patients are homogenous coming from a well-conducted prospective multi-center phase 2 trial with reliable clinical data and no lost of follow-up. All patients received pemetexed-platinum based doublet as first-line treatment with a classical median PFS of 5.75 months. There was as usual a majority of epithelioid subtype (82%), with a majority of stage III disease (71%). Maturity of survival analyses looks pretty good since 100 patients/164 already died (61%) at the time of the analyses, although 114 events were expected for efficacy analysis of the experimental arm. Besides, those are patients with relatively large tumors, at least no amenable to surgery, thus differing from the molecular data mainly deriving form surgical series.

We also found genetic alterations in several genes that might play a role in MPM pathogenesis and progression. The *RDX* gene encodes radixin, a cytoskeletal protein that may be important in linking actin to the plasma membrane. Radixin is a member of the ezrin/radixin/moesin (ERM) family of proteins, which was first isolated as a constituent of adherens junctions in rat liver. A number of studies have suggested that *RDX* is a putative pro-metastatic gene. RDX and moesin are downregulated in some lung adenocarcinomas, suggesting that these two molecules function as tumor suppressors in the early oncogenic stages [14,15].

In particular, the neurofibromatosis 2 (*NF2*) tumor suppressor gene encodes a protein (merlin or schwannomin) structurally related to the ERM family of proteins. Overexpression of the *NF2* gene in NIH3T3 cells decreases their growth rate, confirming the role of *NF2* as a tumor suppressor. Merlin is downregulated in sporadic and *NF2*-related schwannomas.

Inactivating mutations in the *NF2* gene have been reported in 35–40% of MPM patients [16,17].

In our study, *RDX* mutations were more frequent in non-epithelioid histotypes and in the population of patients with asbestos exposure, while *NF2* variants were more frequently found in patients with epithelioid tumors. No significant correlation was found between *RDX* and PFS1. However, although not statistically significant, *RDX* was more frequently mutated in patients with metastatic disease, in agreement with its possible role as a pro-metastatic gene.

The *MXRA5* gene encodes a matrix-remodeling associated protein. This protein contains 7 leucine-rich repeats and 12 immunoglobulin-like C2-type domains related to perlecan. The *MXRA5* protein is aberrantly expressed in non-small-cell lung carcinoma (NSCLC), and a high *MXRA5* expression is correlated with tumor progression and overall survival, indicating its potential value as a novel therapeutic target for the treatment of NSCLC [18]. In our study, *MXRA5* was more frequently mutated in Stage III tumors, epithelioid histotypes, and patients with PFS1 > 6 months.

In our analysis, *BAP1* showed a significant correlation with PFS1, with a possible prognostic role at the first line of treatment with a platinum regime (*p* 0.04). In a recent paper, Hassan et al. [3] evaluated the relationship between OS and platinum chemotherapy. Following platinum chemotherapy, overall survival was significantly longer for patients with loss-of-function mutations in *BAP1* and DNA repair genes compared with patients with no such mutations. The effect of the genotype was highly significant for patients with pleural mesothelioma but not for patients with peritoneal mesothelioma and remained significant after adjusting for gender and age at diagnosis.

Inactivating mutations in *BAP1* can be found in 23% of mesotheliomas [19,20]. Since 2011, over 600 articles have evaluated the role of *BAP1* mutations in mesothelioma and in other cancers. *BAP1* is a deubiquitylase that modulates the activity of multiple genes and proteins controlling DNA replication, DNA repair, metabolism, and cell death. Recent reports have elucidated the mechanism responsible for the potent tumor suppressor activity of *BAP1* [21,22].

*BAP1* germline mutations have been also associated with an increased risk of mesothelioma. In this regard, mesotheliomas in carriers of *BAP1* mutations are almost exclusively of the epithelioid type, are well differentiated, and have an overall nonaggressive morphology, consistent with prolonged survival (i.e., oval cells with bland nuclei, rare mitoses, no necrosis) [23].

We had no available whole blood of non-involved tissue from *BAP1* mutant tumors to assess the germline or somatic origin of *BAP1* mutations. However, the majority of *BAP1* variants had allelic frequency < 50%, suggesting a somatic origin.

Particularly interesting was the correlation between *ACTG1* and *RAPGEF6* and the non-epithelioid histotype, with significant *p* values of 0.027 and 0.035, respectively. Actin (*ACTG1*) is a highly conserved protein that are involved in various types of cell motility and in the maintenance of the cytoskeleton. It has ubiquitous expression in the ovary, esophagus, and 25 other tissues [24]. *RAPGEF 6* (Rap guanine nucleotide exchange factor 6), an activator of the small G-protein Rap1, is a critical regulator of cell–cell contacts and is activated by the remodeling of adherens junctions [25]. Its role in MPM is still poorly studied. Further correlation with OS and PFS could be very interesting for determining its role in the stability of the cell structure.

Other genomic alterations noted in our study regarded *CUL1* and *DHFR.*
*CUL1* is a core component of SCF E3 ubiquitin-protein ligase complexes that mediate the ubiquitination of proteins involved in cell cycle progression, signal transduction, and transcription. A number of studies have reported that CUL1 is a putative pro-metastatic gene, in contrast to other studies [26].

Dihydrofolate reductase (*DHFR*) converts dihydrofolate into tetrahydrofolate, a methyl group shuttle required for the de novo synthesis of purines, thymidylic acid, and certain amino acids. Pemetrexed and its polyglutamated derivatives inhibit thymidylate synthase, dihydrofolate reductase, and glycinamide ribonucleotide transformylase, all of which are involved in the de novo biosynthesis of thymidine and purine nucleotides. Antimetabolite agents, including pemetrexed, induce an imbalance in the cellular nucleotide pool and inhibit nucleic acid biosynthesis, which results in arresting the proliferation of tumor cells and inducing cell death [27,28]. Both *CUL1* and *DHFR* variants have been identified at a high frequency in patients with PFS1 > 6 months and in epithelioid and Stage III tumors, possibly correlating with a good prognosis, in contrast to other scientific evidence [29,30]. In our study, the absence of statistical significance could be caused by the low frequency of these variants (*CUL1* 5%, *DHFR* 1%).

Our data, in line with previous reports, contribute to the characterization of the genetic features of MPM. However, we are aware of the limitations of our study, among which the use of a target gene panel strategy for the genetic profiling. We acknowledge that the use of a wider approach like whole-exome sequencing (WES) would allow a more extensive evaluation of the genetic asset of our study set.

## 6. Conclusions

The results of previous WES studies in MPM demonstrated that MPMs are characterized by a low mutational burden and a low genetic complexity, with very few significantly recurrent mutations. In this regard, our panel, designed on the basis of a careful literature revision, was developed to cover the most frequent alterations in MPM, thus ensuring a significant coverage of genetic events within our cohort. In conclusion, this first analysis of the tumor samples from MPM patients enrolled in the RAMES trial identified relevant prognostic biomarkers that might be useful for patients’ stratification and for the development of novel therapeutic approaches. *BAP1* showed a possible prognostic role at the first line of treatment with platinum/pemetrexed regimens.

The definition of molecular subgroups could be important for identifying groups of patients who could potentially benefit from different strategies of treatment. However, the data will be correlated with survival, experimental treatment response, and liquid biopsies performed during the second-line treatment. This information on a rare tumor type will be important for patients “in real life”.

## Figures and Tables

**Figure 1 cancers-12-02948-f001:**
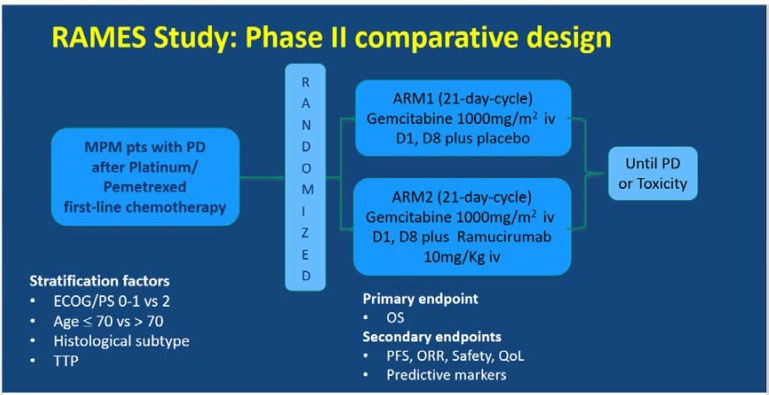
Ramucirumab Mesothelioma clinical trial (RAMES) study design. MPM, malignant pleural mesothelioma, PD, progressive disease, TTP, time to progression, OS, overall survival, PFS, progression-free survival, ORR, overall response rate, QoL, quality of life.

**Figure 2 cancers-12-02948-f002:**
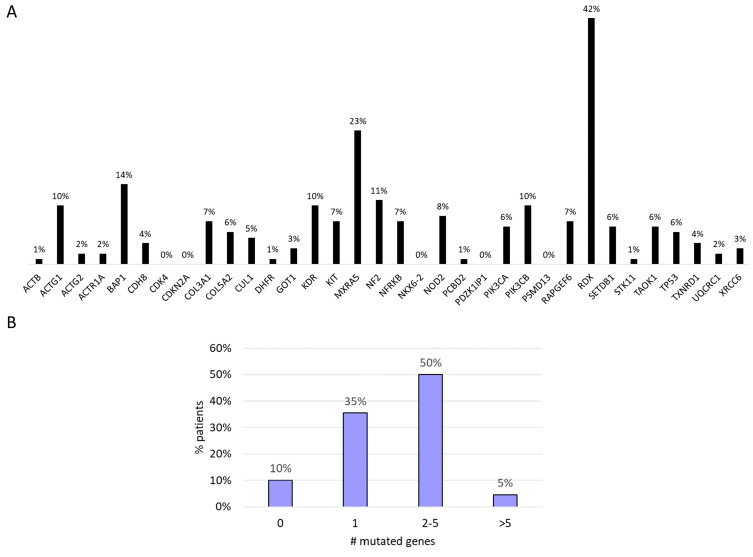
(**A**) Mutational profile. (**B**) Number of gene mutations per patients.

**Figure 3 cancers-12-02948-f003:**
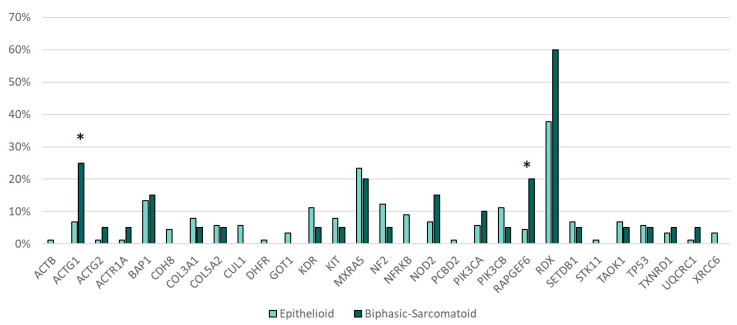
Gene mutations and histotype. * Statistically significant (*ACTG1 p* = 0.027, *RAPGEF6 p* = 0.035).

**Figure 4 cancers-12-02948-f004:**
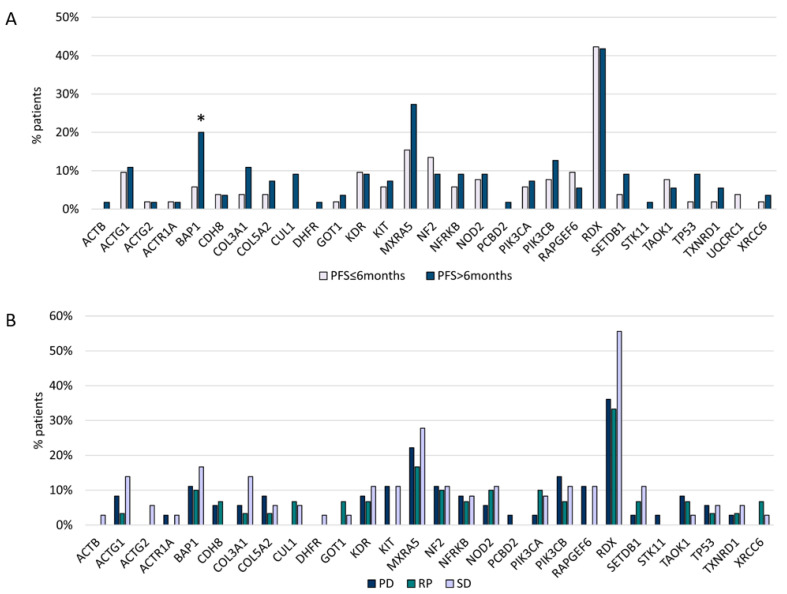
(**A**) Gene mutations and first-line chemotherapy PFS. (**B**) Gene mutations and first-line chemotherapy ORR. * Statistically significant (*BAP1 p* = 0.04).

**Figure 5 cancers-12-02948-f005:**
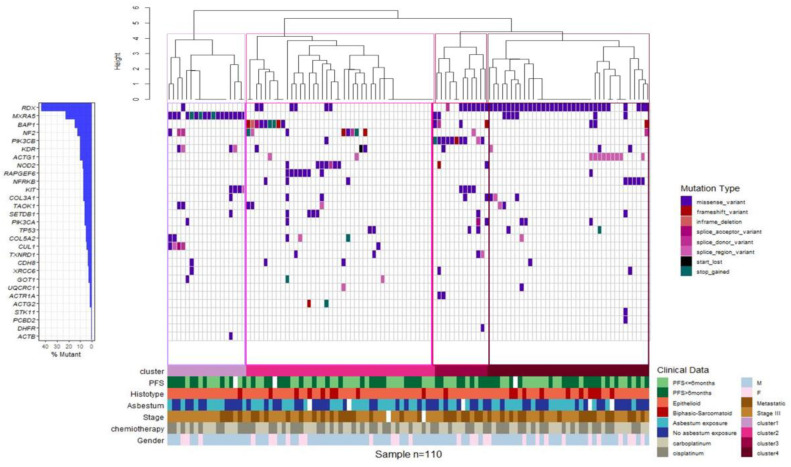
Gene expression-based clustering shows four molecular subgroups. Clustering of MPM cases based on the mutational status. Waterfall plot summarizing the gene variants detected in each patient (columns). Colored squares indicate mutation types. Upper histograms represent for each patient the estimated number of mutations per MB, based on 90.875 bps covered by the panel. Colored line at the bottom indicates patients classified in different genetic clusters.

**Table 1 cancers-12-02948-t001:** Patients’ characteristics (*n* = 110 patients). PR, partial response, SD, stable disease, PD, progressive disease.

Patients’ Characteristics	Results	N (%)
Self-Reported Asbestos Exposure	Exposed	53 (48.2%)
	Not exposed	52 (47.3%)
	Unknown	5 (4.5%)
Sex	Male	82 (74.5%)
	Female	28 (25.5%)
Age	Median	63 (range 45–81)
Histological subtype	Epithelioid	90 (81.8%)
	Biphasic	16 (14.5%)
	Sarcomatoid	4 (3.7%)
Stage	Stage III	68 (71.5%)
	Stage IV	42 (28.5%)
First-Line ORR	PR	30 (27.3%)
	SD	36 (32.7%)
	PD	36 (32.7%)
	Unknown	8 (7.3%)

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
