# Peer review of "Mutational Profile of Malignant Pleural Mesothelioma (MPM) in the Phase II RAMES Study"

_cancers, 2020, doi:10.3390/cancers12102948_

Round 1

Reviewer 1 Report

The manuscript “Mutational profile of malignant pleural mesothelioma (MPM) in the Phase II RAMES Study” by Pagano et al describes a large targeted sequencing study using tumor tissue from patients enrolled in the RAMES Study. The manuscript is clearly written, and the analysis is adequate. Pertinent figures and tables are included. However, the manuscript presents some critical flaws.

Major:

  • Please, describe in more detail the criteria used to call a “mutation” (number of reads, presence of mutation on both strands, etc.).
  • It would be appropriate to validate some of the mutations. In addition, a paragraph about the limitations of the study should be added (i.e. validation of mutations, no normal tissue sequenced, etc.).

Minor:

  • The figures need to be changed with other figures with better resolution
  • The authors report that 48.2% of patients had a history of exposure to asbestos. Is the history documented? In case, it is not (common problem in mesothelioma), it would be more appropriate to indicate it as self-reported asbestos exposure
  • In the discussion, please note that mutations in DHFR have been reported in Bueno, PLOSOne 2009.

Author Response

Reviewer 1 major

Please, describe in more detail the criteria used to call a “mutation” (number of reads, presence of mutation on both strands, etc.).

line 114 Mutation analysis, annotation and filtering were performed using VariantStudio software (Illumina). Mutations were selected based on VCF filters excluding strand bias, low quality and low coverage variants. Only genetic alterations classified as “PASS” in .vcf files were considered. Mutations were considered reliable if sequenced with a minimum coverage of 1000X. Only mutations with a frequency in ExAC general population < 0.1%, described as non-synonymous and non-intron, were considered.

It would be appropriate to validate some of the mutations. In addition, a paragraph about the limitations of the study should be added (i.e. validation of mutations, no normal tissue sequenced, etc.).

DISCUSSION: line 215 We are aware that this study present some technical limitations. Due to the low allelic frequency of the majority of somatic mutations their validation using more standard but less sensitive techniques such as Sanger sequencing is not feasible. Moreover, it was not possible to compare tumor with matched normal counterpart because of limitation on biological material. To overcome this problem a bioinformatic selection based on variants effect and reported frequency in total population was carried out in order to identify somatic alterations.

Can the authors clarified on the selection of the panel of 34 genes detailing  why they do not have explored  some well known genes usually reported in MPM

DISCUSSION line 198 We acknowledge that the choice of a restricted panel of genes is one of the limitations of this work and that the use of a wider approach would have permitted a deeper investigation of RAMES patients genetic profile. However, previous studies conducted on MPM had already demonstrated that this kind of tumor is characterized by low genetic complexity and by the existence of a few number of recurrently mutated genes. For this reason we included in the panel only a selection of genes extracted from a careful literature revision to cover the most frequent alterations in MPM and to ensure a statistically relevant representation of these genetic events within our cohor 11

Reviewer 1 Minor

1-The figures need to be changed with other figures with better resolution ok

2- The authors report that 48.2% of patients had a history of exposure to asbestos. Is the history documented? In case, it is not (common problem in mesothelioma), it would be more appropriate to indicate it as self-reported asbestos exposure..   Confirm self-reported asbestos exposure

           3 In the discussion, please note that mutations in DHFR have been reported in Bueno, PLOSOne 2009…….  bibliography entry not found

Reviewer 2 Report

The authors aim to evaluate the mutational profile of malignant pleural mesothelioma in the phase II RAMSES study and to achieve the correlation between genetic profile observed in 110 MPM patients and  outcomes of platinum/pemetrexed first-line chemotherapy.

This is an interesting paper identifying recent reported genes of interest  in a unique population for evaluation of the genetic landscape.

This article need minor revision before publication

1° Can the authors clarified on the selection of the panel of 34 genes detailing  why they do not have explored  some well known genes usually reported in MPM. 

2°The demographic data, % of histotypes are in agreement with the literature, and the methods of statistical analysis is well known for evaluation. Stronger results could be done in performing a  multivariate analysis  cox model proportional hazards regression to better evaluate  the most relevant genes affecting response to treatment in each arm and survival. 

3° 10% of patients presented no genetic mutations. Is these patients presented a better or a worse response to treatment and what was their survival

4°275 functional somatic mutations were identified in the 34 genes panel.  BAP1 variants was associated with a long PFS after platinum/pemetrexed  first line treament p=0.004. Did the authors found biphasic and sarcomatoid meso BAP1 mutated and is that was also true? 

5° It would be interesting to see some survival curves of the most frequent mutated genes according to  their histotype. and for the evaluation of long responders compared to short survivors

Author Response

Reviewer 2

    1. Can the authors clarified on the selection of the panel of 34 genes detailing  why they do not have explored  some well known genes usually reported in MPM 
    2. DISCUSSION line 198 We acknowledge that the choice of a restricted panel of genes is one of the limitations of this work and that the use of a wider approach would have permitted a deeper investigation of RAMES patients genetic profile. However, previous studies conducted on MPM had already demonstrated that this kind of tumor is characterized by low genetic complexity and by the existence of a few number of recurrently mutated genes. For this reason we included in the panel only a selection of genes extracted from a careful literature revision to cover the most frequent alterations in MPM and to ensure a statistically relevant representation of these genetic events within our cohor 11
  • The demographic data, % of histotypes are in agreement with the literature, and the methods of statistical analysis is well known for evaluation. Stronger results could be done in performing a  multivariate analysis  cox model proportional hazards regression to better evaluate  the most relevant genes affecting response to treatment in each arm and survival
  1. The RAMES study with efficacy analysis on the experimental treatment is still ongoing
  2. 10% of patients presented no genetic mutations. Is these patients presented a better or a worse response to treatment and what was their survival……. Survival data analysis is still ongoing
  3. 275 functional somatic mutations were identified in the 34 genes panel.  BAP1 variants was associated with a long PFS after platinum/pemetrexed  first line treament p=0.004. Did the authors found biphasic and sarcomatoid meso BAP1 mutated and is that was also true? Yes is true
  4. It would be interesting to see some survival curves of the most frequent mutated genes according to  their histotype. and for the evaluation of long responders compared to short survivors…. Survival data analysis is still ongoing

Reviewer 3 Report

The manuscript “Mutational profile of malignant pleural mesothelioma (MPM) in the Phase II RAMES Study” by Pagano et al analyzed the genetic events in samples at diagnosis of patients included in a randomized placebo controlled clinical trial for second line treatment. Using a panel covering the published mutations in MPM, correlations of genetic events with subtypes and PFS after first line therapy with Platinum and Permetrexed were performed. The average number of somatic mutations was associated with PFS and with higher stage. RAPGEF6 and ACTG1 were more frequently associated with sarcomatoid and biphasic as compared to the epitheloid subtype. BAP1 showed a possible prognostic role being associated with PFS. The authors conclude that the information on molecular subgroups could be important information for identifying patients, who could benefit from different treatment strategies.

Suggestions for improvement:

Major improvements

This is a molecular analysis on material of patients with MPM at diagnosis. The results were correlated with histological subtype and progression free survival after first line treatment. Except that the patients were later included in the RAMES Study, there is no real need to include the study in the title or to show the design of the study in a figure. The reader will otherwise be confused by unnecessary information.

Minor improvements

Definition of PFS and how correlations with genetic alterations were performed should be given in detail.

Did the authors perform multivariate analysis to exclude a correlation of molecular findings with interdependent variables subgroups, stage and PFS?

The conclusion should be stronger (e.g. prospective analysis should or will be done)

Author Response

Reviewer 3

Major improvements

  • This is a molecular analysis on material of patients with MPM at diagnosis. The results were correlated with histological subtype and progression free survival after first line treatment. Except that the patients were later included in the RAMES Study, there is no real need to include the study in the title or to show the design of the study in a figure. The reader will otherwise be confused by unnecessary information 
  1. We described the RAMES study to explain the nature of the ongoing study as well as the correlation of genetic data with liquid biopsy and survival data

Minor improvements

  • Definition of PFS and how correlations with genetic alterations were performed should be given in detail.

Line 89 The PFS1 was collected retrospectively and calculated from the date of the 1st cycle to the radiological progression.

The ORR of each treatment was calculated as defined by RECIST v 1.1 guidelines. Not central revision was performed

  • Did the authors perform multivariate analysis to exclude a correlation of molecular findings with interdependent variables subgroups, stage and PFS? 
    1.  
    2. No at the moment, further analyses will be obtained survival data with the two treatment arms of the RAMES study.
  • The conclusion should be stronger (e.g. prospective analysis should or will be done) 
  1. Line 316 However, the data will be correlated with survival, experimental treatment response and liquid biopsy performed during 2-line treatment

Round 2

Reviewer 1 Report

"To overcome this problem a bioinformatic selection based on variants effect and reported frequency in total population was carried out in order to identify somatic alterations."

Many previous studies have validated many mutations in the tumors using Sanger sequencing. It is still not clear how many reads containing the variation were necessary to call a "mutation". Some mutations should have been validated in the original tumors. 

A better literature search would have identified the reference.

https://doi.org/10.1371/journal.pone.0010612

Author Response

Mutations were selected based on VCF quality filters excluding strand bias, low quality and low coverage variants. Mutations were considered “passing filter” if covered by at least 1000 reads and if identified with a minimum frequency of 3% (not less than 40 reads containing the variation). Somatic and functional mutations were identified by a bioinformatic selection that excluded from further analyses variants described with a frequency in ExAC general population higher than 0.1% or classified as synonymous and intronic.

About 75% of identified variants were present with a frequency lower than 20% and their validation using more standard but less sensitive techniques such as Sanger sequencing is not feasible. Aware of the importance of variant calling validation, we are in the process of developing new approaches for the validation of the most functionally relevant variants based on more sensitive techniques like Droplet digital PCR. This activity will take a significant set up time which will fall out of the timing of this manuscript and likely will be the subject of future researches.
